# Antiviral Properties of *Pennisetum purpureum* Extract against Coronaviruses and Enteroviruses

**DOI:** 10.3390/pathogens11111371

**Published:** 2022-11-17

**Authors:** Yi-Ning Chen, Wenny Mei-Wen Kao, Shu-Chi Lee, Jaw-Min Wu, Yi-Sheng Ho, Ming-Kun Hsieh

**Affiliations:** 1Department of Bioscience Technology, Chung Yuan Christian University, Taoyuan 320314, Taiwan; 2Institute of Plant Biology, National Taiwan University, Taipei 10617, Taiwan; 3Graduate Institute of Microbiology and Public Health, National Chung Hsing University, Taichung City 40227, Taiwan

**Keywords:** enterovirus 71, feline coronavirus, infectious bronchitis virus, epidemic diarrhea disease virus, *Pennisetum purpureum*, antiviral activity

## Abstract

Many severe epidemics are caused by enteroviruses (EVs) and coronaviruses (CoVs), including feline coronavirus (FCoV) in cats, epidemic diarrhea disease virus (PEDV) in pigs, infectious bronchitis virus (IBV) in chickens, and EV71 in human. Vaccines and antiviral drugs are used to prevent and treat the infection of EVs and CoVs, but the effectiveness is affected due to rapidly changing RNA viruses. Many plant extracts have been proven to have antiviral properties despite the continuous mutations of viruses. Napier grass (*Pennisetum purpureum*) has high phenolic content and has been used as healthy food materials, livestock feed, biofuels, and more. This study tested the antiviral properties of *P. purpureum* extract against FCoV, PEDV, IBV, and EV71 by in vitro cytotoxicity assay, TCID_50_ virus infection assay, and chicken embryo infection assay. The findings showed that *P. purpureum* extract has the potential of being disinfectant to limit the spread of CoVs and EVs because the extract can inhibit the infection of EV71, FCoV, and PEDV in cells, and significantly reduce the severity of symptoms caused by IBV in chicken embryos.

## 1. Introduction

Coronaviruses (CoVs) and enteroviruses (EVs) have caused severe diseases with high morbidity and mortality in humans and animals [1,2,3,4]. The ongoing COVID-19 pandemic is the most recent outbreak of severe acute respiratory syndrome coronavirus (SARS-CoV)-2. As of August 2022, more than 500 million confirmed cases and 6 million deaths associated with COVID-19 have been confirmed worldwide, and in Taiwan, more than 4 million cases and 9000 deaths associated with COVID-19 have been confirmed. The first COVID-19 outbreak in Taiwan occurred in May 2021, and the second and most severe outbreak occurred in April 2022 [5]. Animal CoVs, such as feline CoV (FCoV), porcine epidemic diarrhea virus (PEDV), and infectious bronchitis virus (IBV), also cause severe diseases with high morbidity and mortality in cats, pigs, and chickens, respectively [6,7,8,9,10]. Feline infectious peritonitis (FIP), which is caused by a virulent strain of FCoV, can lead to high fatality and progressive multi-system disorders in young cat populations [6]. PEDV causes enteric diseases in pigs of all ages, and its clinical manifestations include acute watery diarrhea, dehydration, and vomiting; it leads to high mortality among nursey piglets, which causes tremendous losses in the swine industry worldwide [7,8]. IBV, which was discovered in the 1930s, causes respiratory, reproductive, and renal diseases in chickens of all ages, and has resulted in significant economic losses in the poultry industry [9,10]. Cases of EV infection occur throughout the year in Taiwan. Young children infected with EV may experience complications such as severe neurological diseases or death. The mortality rate is between 1.3% and 33.3%. In Taiwan, the major species in the genus Enterovirus most likely to be accompanied by severe complications is enterovirus 71 (EV71) [3,4].

CoVs have a positive-sense, single-stranded RNA genome that is tightly packed and has a nucleocapsid (N) protein at the center and an outer lipid envelope. Spike (S), membrane (M), and envelope (E) proteins are inserted into the outer lipid envelope. The life cycle of CoVs begins with the binding of the S proteins of CoVs to the receptors on the surfaces of susceptible cells and the subsequent fusion of the viral and cellular membranes. After the entry of CoVs, new viral RNA and proteins are transcribed, translated, and processed by cellular and viral enzymes, including RNA-dependent RNA polymerase (RdRp), 3-chymotrypsin-like protease (3CLpro), and papain-like protease (PLpro) [11]. EV71 is a nonenveloped virus with a positive-sense, single-stranded RNA genome enclosed within a pentameric icosahedral capsid, which encodes a polyprotein that is processed into structural proteins (VP1–VP4) and nonstructural proteins (2A–2C and 3A–3D) by viral proteases (2Apro, 3Cpro, and 3CDpro) [12]. CoVs and EVs are RNA viruses that mutate rapidly. Vaccines and antiviral drugs that are effective against a specific variant may be ineffective against other variants [7,13,14,15]. Herbal extracts are based on antiviral mechanisms different from those of vaccines and antiviral drugs and may be more effective against CoV s and EV constantly emerging variants; thus, herbal extracts are considered viable prophylactic and therapeutic options to reduce the severity of viral diseases [16,17,18]. In addition, the cost of using antiviral drugs in economic animals is too high, and the demand for adding them to feed and drinking water to improve the resistance of animals to CoVs has become increasingly important.

Napier grass Taishigrass (*Pennisetum purpureum* Schumach) is highly efficient in its use of water and nitrogen, and its conversion of light into biomass energy; in addition, it has a high tolerance to a variety of adverse soil conditions, including high salinity and waterlogging; these properties allow the grass to grow in the marginal zone of agriculture cultivation land and yield large quantities of biomass, even in environments with limited resources [19]. The Agriculture Committee of the Executive Yuan has promoted the cultivation of *P. purpureum* as an energy crop [20]. *P. purpureum* is also a food source and can be used in knitting, medicine, papermaking, and biofuel production [21,22]. *P. purpureum* is known for its health benefits and has been used in beverages and food production processing. Studies of feeding, toxicity, gene mutation, hematology, serum chemistry, and pathology in mice have demonstrated that *P. purpureum* is non-toxic. *P. purpureum* extract is rich in polyphenolic compounds, which can help to scavenge free radicals and inhibits peroxidation [23]. Polyphenolic compounds, which are promising inhibitors of viruses in herbal extracts, are small molecules with conjugated fused ring structures and are categorized into flavonoids (flavanols, catechins, anthocyanins) and non-flavonoids (phenolic acids, tannins, stilbenes) [24]. Dietary polyphenols display immunomodulatory capabilities involving inflammation control and immune responses [25]. In vitro experiments have shown that epigallocatechin (EGC), belonging to one of the catechins in the flavonoids of polyphenols in green tea, inhibits the infection of PEDV in Vero cells [26]. The addition of green tea byproducts to the feed and drinking water of chickens is associated with a significant antiviral effect against the H1N1 influenza virus [27]. *P. purpureum*, like green tea, contains a high level of polyphenols and may also have antiviral effects. However, *P. purpureum* can be grown without the use of pesticides and contains only a small amount of caffeine. In addition, as a poultry feed, *P. purpureum* is cheaper than green tea.

We investigated the antiviral properties of the pulverized extract of *P. purpureum* against FCoV, PEDV, IBV, and EV71 to determine whether it could act on both CoVs, which have a lipid envelope, and EVs, which do not have a lipid envelope. The findings may be useful in determining whether *P. purpureum* extract is useful in COVID-19 prevention as the disinfectant of drinking water, feed, and the environment for limiting the spread of viruses in the human disease control, company animal, poultry and swine industries.

## 2. Materials and Methods

### 2.1. Preparation of Pennisetum purpureum Extract (Heyiya^®^)

The anaerobic decomposition reaction method was performed at 600 °C to 700 °C for 1 h to prepare a pulverized crude extract from the fresh fibers separated from the stems and leaves of *P. purpureum*. The *P. purpureum* extract was prepared by HerbRay™ Biotech, Ltd. (Taipei City, Taiwan) and named Heyiya^®^. The pH of diluted extract (1×, 1/2×, 1/4×, 1/8×, 1/10×, 1/100×, 1/200×, 1/300×, and 1/600×) was measured by a pH-009(I) pen type meter (RongZhan).

### 2.2. Determination of Total Phenolic Content

Folin–Cicalteu’s (F-C) phenol reagent (Sigma-Aldrich, St. Louis, MO, USA) was used to determine the total phenolic content (TPC) of the *P. purpureum* extract in the dilution of 1×, 1/2×, 1/4×, 1/8×, 1/10×, 1/100×, 1/200×, 1/300×, and 1/600× according to the method described previously [28]. Briefly, an aliquot of 20 μL samples was incubated with 100 μL of F-C reagent and 80 μL of a 3% Na_2_CO_3_ solution for 20 min. The absorbance values of optical density (OD) were measured at 765 nm using BioTek Synergy multi-detection microplate reader (BioTek, Winoosk, VT, USA), and the concentrations of TPC, which were expressed as mg/L gallic acid equivalent (GAE), were determined by a calibration curve graph (R^2^ = 0.9993).

### 2.3. Cell Lines and Viruses

To test the antiviral properties of the *P. purpureum* extract on the enveloped CoVs and non-enveloped EVs, FCoV serotype II FIPV strain NTU156, PEDV strain Pingtung 52, IBV strain TW-2, and EV-A71 strain 2231 (TW/2231/98) are chosen. *Felis catus* whole fetus-4 (fcwf-4) cells for FCoV were maintained in Minimum Essential Media (MEM). Vero cells for PEDV and Rhabdomyosarcoma (RD) cells for EV71 were maintained in Dulbecco’s Modified Eagle Medium (DMEM). Both media were cultured with 10% fetal bovine serum (FBS), 100 IU/mL penicillin, and 100 IU/mL streptomycin solution in 5% CO_2_ at 37 °C. IBV was propagated in 10-day-old embryonated specific pathogen-free (SPE) chicken eggs (JD-SPF Biotech Co., Ltd., Miaoli, Taiwan) by inoculating the virus into the chorioallantoic sac of eggs. After the propagation and harvest, the extracellular viruses were collected by centrifuging at 500×g for 5 min and the intracellular viruses were released via three frozen-thawed cycles. The virus titer was determined with a 50% tissue culture infectious dose (TCID_50_) assay.

### 2.4. In Vitro Cytotoxicity Assay

Cytotoxicity of the P. purpureum extract was determined by using an MTS assay (CellTiter 96^®^ AQueous One Solution Cell Proliferation Assay, Promega, Madison, WI, USA) for measuring the activity of cellular enzymes that reduce the tetrazolium dye to its insoluble formazan. The assays measured cellular metabolic activity via NAD(P)H-dependent cellular oxidoreductase enzymes and reflect the number of viable cells present. The extract was first adjusted to pH 7 using sterile 1N NaOH and then 10-fold serially diluted to incubate with Fcwf-4 cells or RD cells for 24 h at 37 °C. Three hours after the addition of lysis enzymes into the treated cells to release the color from the cells, OD at 500 nm was measured (BioTek, Winooski, VT, USA). All assays were performed in triplicates. Cell viability percentage was calculated as (1 − (ODtest-ODmedia)/(OD_DMSO_-ODmedia)) × 100%. Cells treated with 100% DMSO were used as the positive control (0% reference), and the cells inoculated with DMEM only served as the negative control (100% reference).

### 2.5. TCID_50_ Assay

RD cells, Fcwf-4 cells, and Vero cells were inoculated to 96-well plates in the concentration of 4 × 10^4^ cells in 100 μL per well and reached a confluent monolayer after 24 h of incubation. Each virus specimen was serially ten-fold diluted into several dilutions. Each dilution was put into 8 wells, and in each well, 100 μL of the diluted virus solution was added. All tests were performed in duplicates. After 3 days of incubation at 37 °C with 5% CO_2_, the infectivity of viruses was determined by observing the formation of cytopathic effects (CPE) and the staining of viable cells by 0.5% crystal violet. The cells were fixed first with methanol for 30 min and then stained with 0.5% crystal violet in methanol for 15 min. The cells stained with 0.5% crystal violet were not infected with viruses, and the infected cells were washed away after the staining. The infectivity percentage was calculated as follows: infectivity% = (number of wells with virus-infected cells/number of wells with virus-inoculated cells) × 100%. The dilution of virus specimen made the cells in four out of eight wells infected is TCID_50_/100 μL, which was calculated as follows: ((infectivity% at the dilution immediately above 50%) – 50%)/(infectivity% at the dilution immediately above 50%) – (infectivity% at the dilution immediately below 50%)).

### 2.6. Virus Inhibition Assay

To determine the antiviral properties of the *P. purpureum* extract, 20 µL of FCoV, PEDV, or EV71 was pretreated with 50 µL of 10-fold, 100-fold, or 1000-fold diluted extract at room temperature for 1 h, 6 h or 12 h. After the designed incubation time, 950 μL of MEM or DMEM cell media was added to each well. Each reaction mixture was adjusted to pH 7 by using sterile 1N NaOH and serially 10-fold diluted and each dilution was put into 8 wells (100 μL/well). All tests were performed in duplicates or triplicates. After 3 days of incubation at 37 °C with 5% CO_2_, the infectivity of viruses was determined by TCID_50_/100 μL. Cells treated with viruses without pretreatment of *P. purpureum* extract were used as the positive control (PC). Relative TCID_50_/100 μL of the virus pretreated with the extract was calculated as follows: log_10_ (TCID_50_/100 μL of the virus without pretreatment of the extract) – log_10_ (TCID_50_/100 μL of virus pretreated with the extract). The inhibition efficacy (inhibition%) was calculated as follows: ((TCID_50_/100 μL of the virus without pretreatment of the extract) – (TCID_50_/100 μL of virus pretreated with the extract))/(TCID_50_/100 μL of the virus without pretreatment of the extract) × 100%.

### 2.7. Antiviral Effect of the P. purpureum Extract against IBV

One-day-old embryonated SPF chicken eggs (JD-SPF Biotech, Miaoli, Taiwan) were kept in an incubator at 38 °C for 9 days, and their viability was checked through candling before tests. To test the toxicity of the P. purpureum extract, the extract diluted in 1×, 1/10×, 1/100×, and 1/1000× was inoculated into 10-day-old embryonated eggs, and no concentrations of the extract were toxic to chicken embryos. Briefly, 20 μL of IBV was pretreated with 50 μL of 10-fold or 100-fold diluted extracts at room temperature for 1 h or 6 h, and then 950 μL of sterile PBS was added to stop the pretreatment. Three-to-five embryonated eggs were inoculated with 100 μL of each reaction mixture (1/10–1 h, 1/10–6 h, 1/100–1 h, 1/100–6 h). Eggs inoculated with IBV without the pretreatment of extract were the IBV control group and the embryos infected with IBV would show the typical symptoms of stunted growth, hemorrhage, or death. Eggs receiving sterile PBS were served as negative control (NC). After 3 days of incubation, all eggs were candled to check embryo viability and then chilled to 4 °C for the necropsy examination of the embryo’s body length and lesions.

### 2.8. Statistical Analysis

Comparisons of results between the two groups were analyzed using an unpaired t-test in the GraphPad Prism 8.0.1 program. The results are expressed as mean ± standard deviation (SD). The *p*-values of <0.05 were regarded as statistically significance, expressed as *, *p* < 0.05; **, *p* < 0.005; ***, *p* < 0.001; and ##, *p* < 0.005.

## 3. Results

### 3.1. Characterization of Pennisetum purpureum Extract

About 300 g of the pulverized crude extract can be acquired from 2 kg of fresh fiber separated from 10 kg of stems and leaves of *P. purpureum*. The extract is acidic and the pH ranged from 5.0 to 5.3. The mean and standard deviation values of TPC were 12,950 ± 755 mg/L GAE calculated in 100-fold to 600-fold dilutions (Table 1).

### 3.2. Cytotoxicity of Pennisetum purpureum Extract (Heyiya^®^)

The toxicity of the *P. purpureum* extract to RD cells and Fcwf-4 cells, which were used to maintain EV71 and FCoV, respectively, was determined by MTS assay. Before pH adjustment, the 100-fold and 1000-fold dilutions of extract were toxic to RD cells and Fcwf-4 cells (Figure 1a and Figure 2a). The low pH of the extract contributed to its cytotoxicity. After the pH of the extract was adjusted to 7, cytotoxicity was observed only in RD cells treated with the 100-fold dilution of extract (Figure 1b and Figure 2b). Thus, the *P. purpureum* extract exhibited stronger cytotoxicity in RD cells than in Fcwf-4 cells.

### 3.3. Antiviral Effect of Pennisetum purpureum Extract (Heyiya^®^) against EV71

After pretreatment of EV71 with 10-fold serial dilutions (10^−2^–10^−4^, log_10_ dilution from 2 to 4) of *P. purpureum* extract for 1 or 6 h at room temperature, RD cells were inoculated with treated EV71 and the antiviral effect of the extract against EV71 was determined by TCID_50_ assay. The 100-fold dilution of *P. purpureum* extract with 6 h of incubation with EV71 achieved the highest virus inhibition rate, reaching 99.96% (***, *p* < 0.001) compared with no treatment of extract and the extract with 1 h of incubation (Figure 3a). Significantly reduced EV71 titer was observed in the EV71 pretreated with 100-fold diluted extract for 6 h compared with the EV71 without pretreatment of the extract (*, *p* < 0.05; Figure 3a). The relative TCID_50_/100 μL was calculated by subtracting Log_10_ TCID_50_/100 μL of the EV71 without the pretreatment of the extract from Log_10_ TCID_50_/100 μL of the EV71 pretreated with the extract. The largest reduction in viral titer was detected in the EV71 pretreated with 100-fold diluted extract for 6 h (Figure 3b). Individual values are presented in Appendix A.

### 3.4. Antiviral Effect of Pennisetum purpureum Extract (Heyiya^®^) against FCoV

After pretreatment of FCoV with 10-fold serial dilutions (10^−1^–10^−3^, log_10_ dilution from 1 to 3) of *P. purpureum* extract for 1 or 6 h at room temperature, Fcwf-4 cells were inoculated with treated FCoV and the antiviral effect of the extract against FCoV was determined by TCID_50_ assay. As compared with the viral titer of FCoV without pretreatment of the extract, a significant reduction in viral titers was observed in the FCoV pretreated with 10-fold and 100-fold diluted extract for 1 h (**, *p* < 0.005) or 6 h (##, *p* < 0.005, Figure 4a, Appendix A). The relative TCID_50_/100 μL was calculated by subtracting Log_10_ TCID_50_/100 μL of the FCoV without the pretreatment of the extract from Log_10_ TCID_50_/100 μL of the FCoV pretreated with the extract. A concentration-dependent reduction in the relative TCID_50_/100 μL was exhibited from the FCoV pretreated with the extract from 10-fold to the 1000-fold dilution (Figure 4b). Individual values are presented in Appendix A.

### 3.5. Antiviral Effect of Pennisetum purpureum Extract (Heyiya^®^) against PEDV

After pretreatment of PEDV with 10-fold serial dilutions (10^−1^–10^−3^, log_10_ dilution from 1 to 3) of *P. purpureum* extract for 1 or 6 h at room temperature, Vero cells were inoculated with treated PEDV and the antiviral effect of the extract against PEDV was determined by TCID_50_ assay. As shown in Figure 5, the infection of the PEDV pretreated with the 10-fold dilution of *P. purpureum* extract for 6 h was completely (100%) inhibited in VERO cells (*p* < 0.005). Furthermore, the infection of the PEDV pretreated with a 10-fold dilution of extract for 1 h was inhibited by 90% in VERO cells (*p* < 0.05). After 6 h of pretreatment with the 1000-fold dilution of extract, the viral titer of PEDV significantly decreased (*p* < 0.05). Individual values are presented in Appendix A.

### 3.6. Antiviral Effect of Pennisetum purpureum Extract (Heyiya^®^) against IBV

After chicken eggs inoculated with IBV pretreated with 10-fold and 100-fold dilutions of P. alopecuroides extract for 1 or 6 h, the antiviral effect of the extract against the IBV TW2 strain was determined on the basis of the severity of clinical signs caused. The typical symptoms of IBV infection in 10-day-old chicken embryos included death, hemorrhage, and stunted growth (Figure 6a). The body lengths of the chicken embryos were measured to ascertain the severity of stunted growth. The pretreatment of *P. purpureum* extract can reduce the severity of IBV disease because the body lengths of the chicken embryos inoculated with IBV pretreated with the 10-fold or 100-fold dilution of *P. purpureum* extract for 1 h or 6 h were significantly greater than those of the embryos inoculated with IBV without pretreatment of the extract (Figure 6b). Individual values are presented in Appendix A. The effects of *P. purpureum* extract were concentration and time-dependent because the chicken embryos inoculated with IBV pretreated with the 10-fold diluted extract for 6 h had the greatest body length in addition to the chicken embryo without the inoculation of IBV (NC).

## 4. Discussion

The use of disinfectants has become commonplace worldwide because of the COVID-19 pandemic. Alcohol and other chemical-based sanitizers recommended by the World Health Organization and United States Food and Drug Administration limit the spread of SARS-CoV-2, other CoVs, EVs, and many other viruses [29,30], but prolonged and excessive use of these sanitizers cause skin damage that facilitates the entry of harmful microbes [31,32]. Many plant-derived compounds have antimicrobial properties and are biodegradable and non-toxic; therefore, they may be useful as disinfectants [33]. Herbal extracts with antiviral properties can be added to feed and drinking water to reduce the spread of viruses as well as morbidity and disease severity [27]. In this study, pretreatment with the pulverized crude extract of *P. purpureum* at room temperature significantly reduced the infectivity of EV71, FCoV, and PEDV and the severity of symptoms caused by IBV in embryonated chickens, demonstrating the potential of the extract as an environmental disinfectant and sterilizing feed additive.

The most salient difference between the structures of CoVs and EVs is the outer membrane of the lipid envelope, which is derived from the cell membranes of previously infected cells. Alcohol-based disinfectants and many chemical-based disinfectants have amphiphilic properties that facilitate access to the viral lipid membrane through protein denaturation and disruption of the lipid envelope [33,34,35]. Such disinfectants are ineffective against EVs because EVs do not have a lipid envelope. However, *P. purpureum* extract was effective against both enveloped CoVs and non-enveloped EVs. The high TPC of the *P. purpureum* extract may explain its antiviral properties because polyphenols can use a variety of antiviral mechanisms in addition to disrupting the structure of the viral lipid envelope. According to the report on the website maintained by the Livestock Research Institute of the Agricultural Committee of the Executive Yuan (Taiwan), the *P. purpureum* extract contained 51 mg/g of flavonoids out of 120 mg/g of total phenols (42%) [21]. The phytochemical screening test conducted by the University of Port Harcourt, Nigeria, revealed that *P. purpureum* contained high levels of tannins, flavonoids, saponins, and alkaloids, known for their antimicrobial activities [36]. Many bioactive compounds categorized as flavonoids and non-flavonoids of polyphenols have been shown potent antiviral activity against CoVs and EVs in vitro, in vivo, and in silico findings [16,17,18], such as catechins of flavonoids against SARS-CoV [37] and PEDV [26], resveratrol (subclass stilbenes of non-flavonoids) against MERS-CoV [38] and EV71 [39], quercetin (subclass flavonols of flavonoids) against SARS-CoV-2 [40] and EV71 [41], and rosmarinic acid (subclass phenolic acids of non-flavonoids) as a pan-coronaviral main proteinase inhibitor [42] and an inhibitor against EV71 [43,44]. A recent study also showed that tannic acid-chelated zinc supplementation in the diet of newborn piglets could alleviate PEDV-induced damage of the intestinal mucosa and improve the absorptive function and growth in piglets [45]. Experimentally validated studies have found polyphenols can prevent the entry of viruses by reducing the levels of viral surface proteins to interfere with the binding of viruses with their cellular receptors [39,40,44,46]; interfere with the release of viral genomes into the host cells by inhibiting the production in viral enzymes associated with membrane fusion and viral uncoating [46,47,48,49] or interacting with viral proteins related to viral disassembly [41,43]; prevent viral replication and transcription by inhibiting the production of viral nucleocapsid proteins [37,38,50], and RNA polymerase [51,52]. Further studies are required to clarify the anti-viral mechanisms used by the *P. purpureum* extract to inhibit the infection of enveloped CoVs and non-enveloped EVs.

*P. purpureum* is the highest yielding forage species in Taiwan according to the Agriculture Committee of the Executive Yuan and is used as livestock feed, an alternative to coal and wood chip fuels, raw material for pulp, soil for mushroom cultivation, and energy biomass with high CO_2_ fixation capacity [22]. The Animal Testing Institute and Nuclear Research Institute have developed cellulosic alcohol with an alcohol–gasoline blend ratio of 3%, which can be used directly by most vehicles [20]. In addition, *P. purpureum* can be fermented with lactic acid bacteria to produce fibrous lactic acid, which can be used as industrial raw material and further synthesized into polylactic acid (PLA) particles for use in bioplastics. PLA is widely used in high-quality biomedical materials, packaging, films, fibers, plastic components of electronic goods, and 3D printing [20,22]. The *P. purpureum* extract (Heyiya^®^) tested in this study was obtained from the solid waste byproducts of such applications. Our results demonstrate the antiviral potential of *P. purpureum* extract (Heyiya^®^) and its applicability to the fields of animal husbandry, agriculture, veterinary medicine, and biomedicine for the control and prevention of the spread of viruses.

## 5. Conclusions

The infectivity of FCoV, PEDV, IBV, and EV71 significantly declined after the pretreatment of *P. purpureum* extract (Heyiya^®^) (Figure 7).

## Figures and Tables

**Figure 1 pathogens-11-01371-f001:**
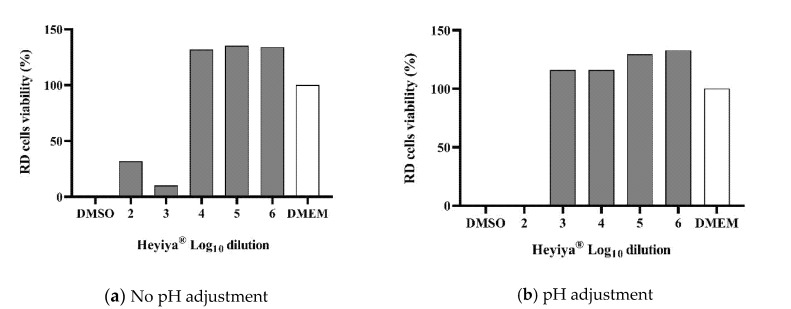
Cytotoxicity of *Pennisetum purpureum* extract (Heyiya^®^) in RD cells determined by MTS assay: (**a**) RD cells were treated with 10-fold serial dilutions of original extract (pH 5~5.3). (**b**) RD cells were treated with 10-fold serial dilutions of pH-adjusted extract (pH 7). The dilutions tested starting from 10^−2^ to 10^−6^, expressed as log_10_ dilution 2 to 6. All assays were performed in triplicate. Cell viability percentage was calculated as (1 − (OD_test_ − OD_media_)/(OD_DMSO_ − OD_media_)) × 100%. Cells treated with 100% DMSO were used as the positive control (0% reference) and cells inoculated with medium DMEM only served as the negative control (100% reference).

**Figure 2 pathogens-11-01371-f002:**
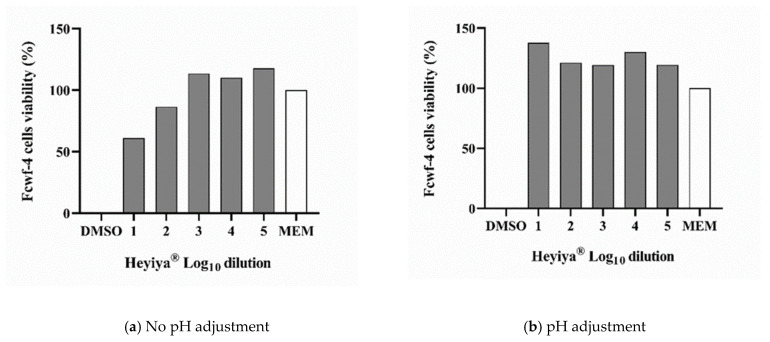
Cytotoxicity of *Pennisetum purpureum* extract (Heyiya^®^) in Fcwf-4 cells determined by MTS assay: (**a**) Fcwf-4 cells were treated with 10-fold serial dilutions of original extract (pH 5–5.3). (**b**) Fcwf-4 cells were treated with 10-fold serial dilutions of pH-adjusted extract (pH 7–7.2). The dilutions tested starting from 10^−2^ to 10^−6^, expressed as log_10_ dilution 2 to 6. All assays were performed in triplicate. Cell viability percentage was calculated as (1 − (OD_test_ − OD_media_)/(OD_DMSO_ − OD_media_)) × 100%. Cells treated with 100% DMSO were used as the positive control (0% reference) and cells inoculated with medium MEM only were served as the negative control (100% reference).

**Figure 3 pathogens-11-01371-f003:**
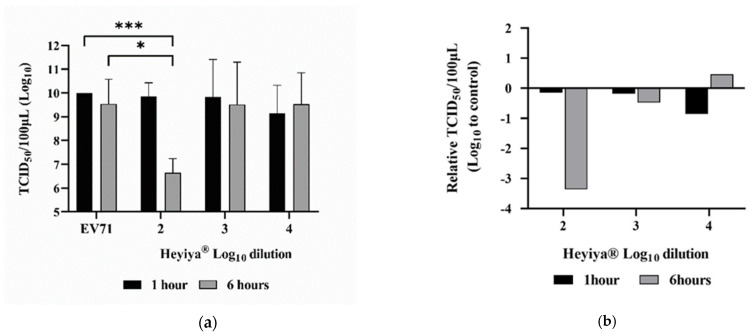
Antiviral effect of *Pennisetum purpureum* extract (Heyiya^®^) against enterovirus 71 (EV71): (**a**) Titers (TCID_50_/100 μL) of the EV71 pretreated with 10-fold serial dilutions of the extract starting from 10^−2^ to 10 ^−4^ (log_10_ dilution from 2 to 4) for 1 h (black) and 6 h (grey) were determined by TCID_50_ assay. The EV71 without pretreatment of the extract was used as the positive control (EV71). Bars indicate means and error bars indicate standard deviation of triplicate experimental data. Comparisons were performed through unpaired t-tests in GraphPad Prism 8.0.1. The *p*-values of <0.05 were regarded as statistically significant, expressed as *, *p* < 0.05; ***, *p* < 0.001. (**b**) Relative TCID_50_/100 μL of EV71 pretreated with the extract was calculated as follows: log_10_ (TCID_50_/100 μL of the EV71 without pretreatment of the extract)-log_10_ (TCID_50_/100 μL of the EV71 pretreated with the extract), plotted against the log_10_ dilutions from 2 to 4 of the extract. The largest reduction in viral titer was detected in the EV71 pretreated with 100-fold diluted extract for 6 h.

**Figure 4 pathogens-11-01371-f004:**
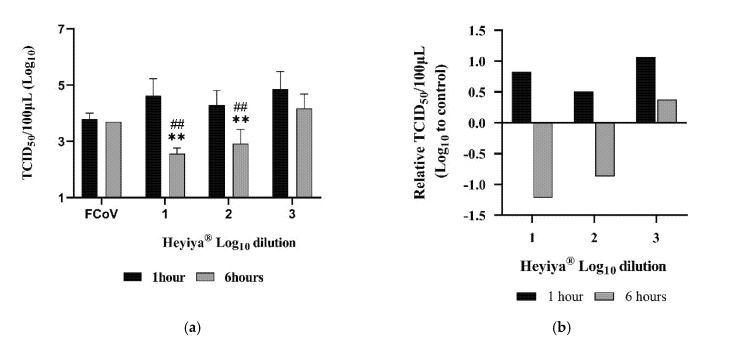
Antiviral effect of *Pennisetum purpureum* extract (Heyiya^®^) against feline coronavirus (FCoV): (**a**) Titers (TCID_50_/100 μL) of the FCoV pretreated with 10-fold serial dilutions of the extract starting from 10^−1^ to 10^−3^ (log_10_ dilution from 1 to 3) for 1 h (black) and 6 h (grey) were determined by TCID_50_ assay. The FCoV without pretreatment of the extract was used as the positive control (FCoV). Bars indicate means and error bars indicate standard deviation of triplicate experimental data. Comparisons were performed through unpaired t-tests in GraphPad Prism 8.0.1. Statistical significance was indicated by a *p*-value of <0.05. “**, *p* < 0.005” indicated the comparisons were between the FCoV without pretreatment of the extract and the FCoV pretreated with 10-fold or 100-fold of the extract for 6 h. “##, *p* < 0.005” indicated the comparisons were between the FCoV without pretreatment of the extract for 1 h and the FCoV pretreated with 10-fold or 100-fold of the extract for 6 h. (**b**) Relative TCID_50_/100 μL of FCoV pretreated with the extract was calculated as follows: log_10_ (TCID_50_/100 μL of the FCoV without pretreatment of the extract)-log_10_ (TCID_50_/100 μL of the FCoV pretreated with the extract), plotted against the log_10_ dilutions from 1 to 3 of the extract. The reduction in viral titer was in concentration-dependent manner.

**Figure 5 pathogens-11-01371-f005:**
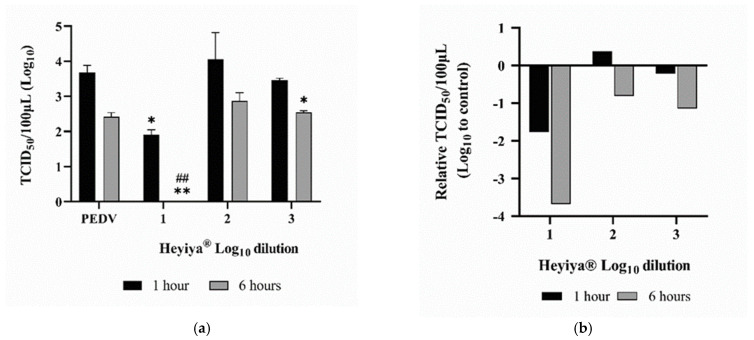
Antiviral effect of *Pennisetum purpureum* extract (Heyiya^®^) against porcine epidemic diarrhea virus (PEDV): (**a**) Titers (TCID_50_/100 μL) of the PEDV pretreated with 10-fold serial dilutions of the extract starting from 10 ^−1^ to 10 ^−3^ (log_10_ dilution from 1 to 3) for 1 h (black) and 6 h (grey) were determined by TCID_50_ assay. The PEDV without pretreatment of the extract was used as the positive control (PEDV). Bars indicate means and error bars indicate standard deviation of duplicate experimental data. Comparisons were performed through unpaired t-tests in GraphPad Prism 8.0.1. Statistical significance was indicated by a *p*-value of <0.05. “*, *p* < 0.05” indicated the comparisons were between the PEDV without pretreatment of the extract for 1 h and the PEDV pretreated with the 10-fold dilution of extract for 1 h, or the PEDV pretreated with the 1000-fold dilution of extract for 6 h. “**, *p* < 0.005” indicated the comparisons were between the PEDV without pretreatment of the extract for 1 h and the PEDV pretreated with 10-fold dilution of extract for 6 h. “##, *p* < 0.005” indicated the comparisons were between the PEDV without pretreatment of the extract for 6 h and the PEDV pretreated with 10-fold dilution of extract for 6 h. (**b**) Relative TCID_50_/100 μL of PEDV pretreated with the extract was calculated as follows: log_10_ (TCID_50_/100 μL of the PEDV without pretreatment of the extract)-log_10_ (TCID_50_/100 μL of the PEDV pretreated with the extract), plotted against the log_10_ dilutions from 1 to 3 of the extract.

**Figure 6 pathogens-11-01371-f006:**
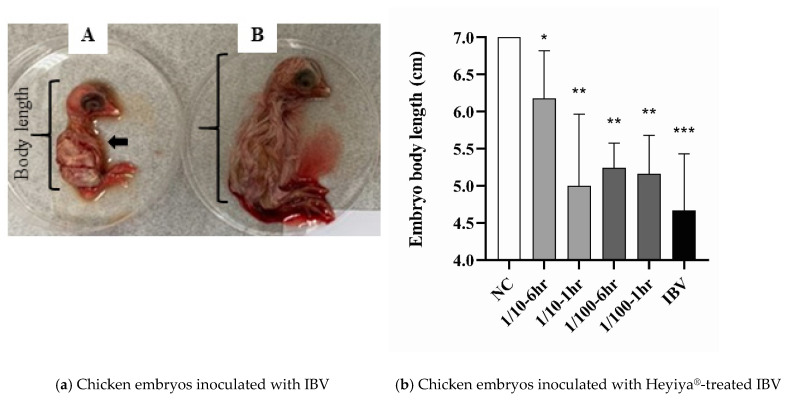
Antiviral effect of *Pennisetum purpureum extract (Heyiya^®^)* against infectious bronchitis virus (IBV): (**a**) Chicken embryos inoculated with IBV (A) had symptoms of hemorrhage (black arrow) and stunted growth with a lesser body length by comparing to the body length of chicken embryos without IBV inoculation (B). (**b**) The body lengths of chicken embryos without IBV inoculation (NC), inoculated with IBV without the pretreatment of *P. purpureum* extract (IBV), and inoculated with IBV pretreated with 10-fold or 100-fold diluted extract for 1 h or 6 h at room temperature (1/10–6 h, 1/10–1 h, 1/100–6 h, 1/100–1 h) were measured and compared through unpaired t-tests in GraphPad Prism 8.0.1. Statistical significance was indicated by a *p*-value of <0.05. “*, *p* < 0.05; **, *p* < 0.005; ***, *p* < 0.001” indicated the comparisons between NC to other groups, respectively.

**Figure 7 pathogens-11-01371-f007:**
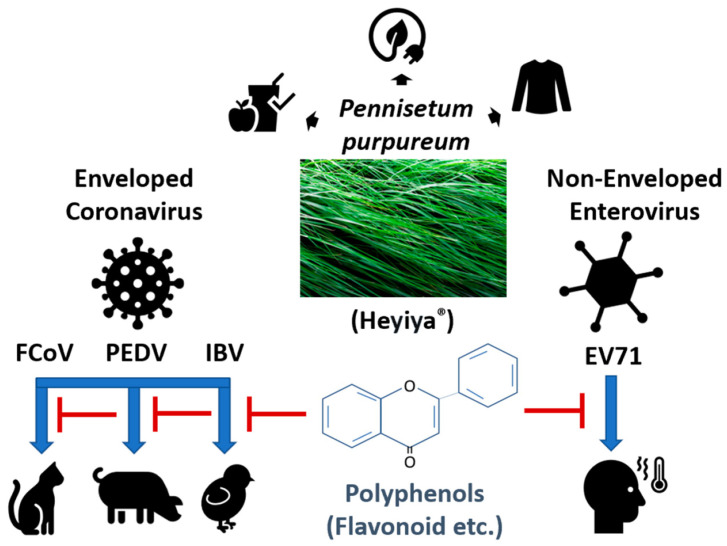
Illustration of antiviral properties of *Pennisetum purpureum* extract (Heyiya^®^) against coronaviruses and enterovirus.

**Table 1 pathogens-11-01371-t001:** The pH and total phenolic content of *Pennisetum purpureum* extract.

Extract Dilution	pH	Total Phenolic Content (TPC) mg/L Gallic Acid Equivalent (GAE)
1×	5.3	1413.708
2×	5.3	1328.513
4×	5.3	1315.921
8×	5.0	984.228
10×	5.1	916.995
100×	5.1	128.490
200×	5.1	66.128
300×	5.1	39.936
600×	5.0	23.188

The concentrations of TPC were determined by a calibration curve (y = 2.3374 × + 0.0767, R^2^ = 0.9993) using Folin–Ciocalteu’s phenol reagent, expressed as mg/L GAE.

## Data Availability

Not applicable.

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
