# Peer review of "Antiviral Properties of Pennisetum purpureum Extract against Coronaviruses and Enteroviruses"

_pathogens, 2022, doi:10.3390/pathogens11111371_

Round 1
Reviewer 1 Report
Manuscript written by Yi-Ning Chen * , Wenny Mei-Wen Kao , Shu-Chi Lee , Jaw-Min Wu , Yi-Sheng Ho , And Ming-Kun Hsieh present experimental study to asses the anti -viral properties some plants extract for using in veterinary to prevent wide spread of infectious infectious bronchitis virus (IBV) in chickens.Napier grass (Pennisetum purpureum) has high phenolic content and has been used as healthy food materials, livestock feed, biofuels, and more. This study tested the antiviral properties of P. purpureum extract against FCoV, PEDV, IBV, and EV71 by in vitro cytotoxicity assay.Current study provides a novel direction in veterinary medicine to prevent a poultry stock from wide spread of diseases caused by coronaviruses and enteroviruses and present a great interest in development of novel antiviral therapy.
Manuscript is written well but some improvements in style would be appropriate, description of assays is detailed. Authors described the limitations of studies and also proposed future directions of antiviral mechanisms of plant extract used in current study such as an entry assay for used viruses.
All results are presented in manuscript described sufficiently ,all conclusions are valid.
I have some specific comments in presentation of results:
1. Use more appropriate toxic control in MTS toxicity study such as detergents or staurosporine not DMSO.
2. Present individual values in Figure 3A,4A,5A, 6B
Author Response
"Please see the attachment."

Reviewer 2 Report
The authors performed a good experiment that demonstrates the antiviral properties of Pennisetum purpureum extract against coronaviruses and enteroviruses. The materials and methods were performed accurately. However, I recommend the authors make some minor improvements.
In the Introduction section, I recommend replacing reference [24] with a newly published one on flavonoids and viruses (DOI: 10.3390/v14030592).
Also, I recommend the authors discuss the potential compounds that might be accountable for inducing the antiviral activities of Pennisetum purpureum extract.
Author Response
Response to Reviewer 2 Comments
Point 1: In the Introduction section, I recommend replacing reference [24] with a newly published one on flavonoids and viruses (DOI: 10.3390/v14030592).
Response 1: We thank the reviewer for the comment. In the Introduction section, reference [24] presents a detailed information about the bioactive components of polyphenoles, including flavonoids. The recommeded reference (Flavonoids Target Human Herpesviruses That Infect the Nervous System: Mechanisms of Action and Therapeutic Insights) has reviewed the mechanisms flavonoids used to target human herpesviruses, which are DNA viruses, and our study is about the antiviral properties of Pennisetum purpureum extract against RNA viruses, Coronaviruses and Enteroviruses. Therefore, we prefer not to replace reference [24] with the recommended reference after thorough consideration.
Point 2: Also, I recommend the authors discuss the potential compounds that might be accountable for inducing the antiviral activities of Pennisetum purpureum extract.
Response 2: According to the reviewer’s recommendation, we added more information about the components of P. purpureum and their antiviral activities in the following sentences in the section of “Discussion”.
- The phytochemical screening test conducted by University of Port Harcourt, Nigeria, showed that P. purpureum contained high levels of tannins, flavonoids, saponins, and alkaloids, known for their antimicrobial activities [Okaranye 2009].
- Recent study also showed that tannic acid-chelated zinc supplementation in the diet of newborn piglet could alleviate PEDV induced damage on intestinal mucosa and antioxidative capacity, and improve the absorptive function and growth in piglets [Zhang 2022].
References:
- Okaraonye, C.C.; Ikewuchi, J.C. Nutritional and antinutritional components of Pennisetum purpureum (Schumach). Pak J Nutr 2009, 8, 32-34. doi: 10.3923/pjn.2009.32.34.
- Zhang, Z.; Wang, S.; Zheng, L.; Hou, Y.; Guo, S.; Wang, L.; Zhu, L.; Deng, C.; Wu, T.; Yi, D.; Ding, B. Tannic acid-chelated zinc supplementation alleviates intestinal injury in piglets challenged by porcine epidemic diarrhea virus. Front Vet Sci 2022, 9, 1033022. doi: 10.3389/fvets.2022.1033022.
Reviewer 3 Report
The paper discusses antiviral properties of Pennisetum purpureum extract against coronaviruses and enteroviruses. An article is well formulated and informative. The main aim of the study is clearly defined. „Introduction” section is well prepared and fully explains to the reader the state of knowledge concerning coronaviruses and enteroviruses issues, as well as broad spectrum of Pennisetum purpureum beneficial properties. Author cites a few important references to support a statement, including COA, FDA and WHO reports related to the topic. The results are clearly presented in the form of one table and 6 figures. Unfortunately, author hasn’t included illustrations. References are relevant and referenced correctly. “Conclusions” section provides the general conclusion, but is a bit too short.
Minor remarks:
1. There is a need to check the manuscript in terms of typos (e. g. page 8 line 317).
2. At least one illustration is needed to improve attractiveness of the paper.
3. Is Pennisetum purpureum able to overcome resistance resulting from the formation of viral RNA mutations?
4. Does the Autor make plans to investigate specific molecular mechanism of Pennisetum purpureum’s action?
Author Response
"Please see the attachment."
